# Pramlintide: A Novel Therapeutic Approach for Osteosarcoma through Metabolic Reprogramming

**DOI:** 10.3390/cancers14174310

**Published:** 2022-09-02

**Authors:** Yuanzheng Yang, Zhanglong Peng, Elsa R. Flores, Eugenie S. Kleinerman

**Affiliations:** 1Department of Pediatrics Research, Division of Pediatrics, The University of Texas MD Anderson Cancer Center, Houston, TX 77030, USA; 2Department of Molecular Oncology, H. Lee Moffitt Cancer Center and Research Institute, Tampa, FL 33612, USA; 3Department of Cancer Biology, The University of Texas MD Anderson Cancer Center, Houston, TX 77030, USA

**Keywords:** osteosarcoma, pramlintide, IAPP, p53, p63, p73, glycolytic metabolism

## Abstract

**Simple Summary:**

Outcomes for patients with osteosarcoma have remained stagnant for more than 3 decades. Novel therapeutic agents are desperately needed. Osteosarcoma cells often express abnormal form of the p53, p63, and p73 genes, which promote glucose uptake, cell glycolysis, and rapid proliferation. Pramlintide, an FDA-approved drug for type 2 diabetes, interfered with tumor glycolysis leading to decreased cell growth and increased cell death. When we injected Pramlintide into osteosarcoma tumor nodules of the mice, the tumors were significantly smaller after 21 days while the control tumors continued to grow. Tumor hypoxia was also decreased. This is the first report showing the potential efficacy of Pramlintide against osteosarcoma, indicating that Pramlintide may be a novel therapeutic approach for patients with relapsed osteosarcoma.

**Abstract:**

Despite aggressive combination chemotherapy and surgery, outcomes for patients with osteosarcoma have remained stagnant for more than 25 years, and numerous clinical trials have identified no new therapies. p53 deletion or mutation is found in more than 80% of osteosarcoma tumors. In p53-deficient cancers with structurally altered p63 and p73, interfering with tumor cell metabolism using Pramlintide (an FDA-approved drug for type 2 diabetes) results in tumor regression. Pramlintide response is mediated through upregulation of islet amyloid polypeptide (IAPP). Here, we showed that osteosarcoma cells have altered p63, p73, and p53, and decreased IAPP expression but have the two main IAPP receptors, CalcR and RAMP3, which inhibit glycolysis and induce apoptosis. We showed that in osteosarcoma cells with high- or mid-range glycolytic activity, Pramlintide decreased cell glycolysis, resulting in decreased proliferation and increased apoptosis in vitro. In contrast, Pramlintide had no effect in osteosarcoma cells with low glycolytic activity. Using a subcutaneous osteosarcoma mouse model, we showed that intratumoral injection of Pramlintide-induced tumor regression. Tumor sections showed increased apoptosis and a decrease in Ki-67 and HIF-1α. These data suggest that in osteosarcoma cells with altered p53, p63, and p73 and a high glycolytic function, Pramlintide therapy can modulate metabolic programming and inhibit tumor growth.

## 1. Introduction

Cure rates for osteosarcoma using multi-agent chemotherapy have remained stagnant for more than 25 years [1,2,3,4,5]. Altering chemotherapy, including dose intensification, has not improved the survival rate beyond 60% to 65%. Survival for patients with recurrent, metastatic, or therapy-refractory disease is less than 20%, and no new effective therapies have been identified in more than two decades. Drugs tested in seven recent Children’s Oncology Group phase 2 trials were deemed ineffective, showing a median time to progression of only 4 months [6]. Genomic analysis has been unsuccessful in identifying consistent targetable options, highlighting the need for novel therapeutic approaches. Pediatric, adolescent, and young adult osteosarcoma patients have also not benefited from the medical breakthroughs in immunotherapy seen in patients with melanoma and lung cancer, despite evidence of the expression of immune checkpoint receptors such as PD-L1 on osteosarcoma cells and patient tumor specimens. To date, the trials with immune checkpoint inhibitors in patients with metastatic osteosarcoma have been disappointing [7,8].

Osteosarcoma tumors have complex genetic alterations with complex chromosome duplication and gene rearrangement that differ from patient to patient and even within multiple metastatic lesions in the same patient [9]. This has made molecularly targeted precision therapy challenging. Therefore, treating patients with relapsed, metastatic, or refractory disease remains imprecise with no consistent optimal strategies at the present time [10,11]. The one constant genetic alteration in osteosarcoma is the deletion or mutation of the TP53 gene, a genetic abnormality seen in more than 50% of human cancers [12,13]. TP53 deletions and mutations have also been associated with Li-Fraumeni syndrome, and patients with Li-Fraumeni tend to develop osteosarcoma [14,15]. The p53 family, including p53 itself, p63, and p73, regulates the expression of a variety of target genes implicated in DNA repair, the induction of cell cycle arrest, cell senescence, and apoptosis [16,17]. In the ~80% of osteosarcoma tumors with TP53 mutations, more than 50% have structural mutations or translocations that result in p53 inactivation [18,19]. Reactivation of p53 has been shown to suppress tumor growth in vivo in mice [20,21]. Unfortunately, despite the uniformity of p53 abnormalities in the majority of patients with osteosarcoma, this strategy of focusing on p53 has proven to be difficult to implement therapeutically.

Recently, Venkatanarayan et al. demonstrated that the deletion of ΔN isoforms of p63 or p73 resulted in metabolic reprogramming of p53-deficient cells and regression of thymic lymphomas in mice through the upregulation of islet amyloid polypeptide (IAPP), a gene that encodes for amylin [22,23]. Amylin is an amino acid peptide that is co-secreted with insulin by the β-cells of the pancreas and has been shown to be deficient in patients with type 1 diabetes [24,25]. Amylin is a glycolysis regulator that reduces glucose blood levels [25]. Upregulating IAPP and increasing amylin inhibited tumor glycolysis and increased reactive oxygen species (ROS) production, leading to tumor cell apoptosis and tumor regression [22]. This anti-tumor effect was also induced by treating the mice with Pramlintide, an FDA-approved synthetic analogue of amylin that is used to treat type 1 and 2 diabetes in conjunction with insulin [26,27]. Pramlintide inhibits secretion of glucagon and lowers blood glucose. Glucose and glucagon are critical in cancer-cell metabolism, and their uptake by cancer cells is enhanced through oncogenic signaling. Tumor cells rely on glycolysis to meet their metabolic demands and for the synthesis of nucleotides and other molecules required for rapid cancer-cell growth. Therefore, therapies that inhibit glucose metabolism and limit or decrease the availability of glucose in the blood may have therapeutic implications for cancers with p53 deletions or mutations, such as osteosarcoma. Due to Pramlintide’s ability to decrease blood glucose, we investigated the in vitro effect of Pramlintide on osteosarcoma cell viability, proliferation, and metabolism and its in vivo effect as a therapeutic approach.

## 2. Materials and Methods

### 2.1. Cell Lines and Reagents

Human osteosarcoma cell lines SAOS-2 (HTB-85), U2OS (HTB-96), 143B (CRL-8303), SJSA-1 (CRL-2098), and MG63 (CRL-1427); human lung adenocarcinoma cells (H1299); human fetal osteoblastic cells (hFOB; CRL-11372); and 293T cells were obtained from the American Type Culture Collection (Manassas, VA, USA). The metastatic LM7 cell line was derived from SAOS-2 cells in our laboratory. The metastatic MG63.2 subline was derived in a similar way from MG63 cells. CCH-OS-O and CCH-OS-D cells were derived from primary patient tumor pretreatment biopsy specimens taken at the Children’s Cancer Hospital, The University of Texas MD Anderson Cancer Center [28]. CCH-OS-D-luc was generated from CCH-OS-D and transfected with luciferase-expressing MIGR1 retrovirus.

All cells except SJSA-1, H1299, and hFOB cell lines were maintained in Dulbecco’s modified Eagle’s medium supplemented with 2 mmol/L L-glutamine, 1 mmol/L sodium pyruvate, 1× nonessential amino acids, 1× minimal essential medium vitamin solution, 10% heat-inactivated fetal bovine serum, 100 U/mL penicillin, and 100 mg/mL streptomycin at 37 °C in 5% CO_2_. The SJSA-1 cells and H1299 cell lines were cultured with RPMI-1640 medium supplemented with 10% fetal bovine serum, 1% L-glutamine, and 1% penicillin/streptomycin. The hFOB cells were maintained in Dulbecco’s minimal essential medium/Ham’s F12 (DMEM/F12-1:1 mix) containing 10% (*v*/*v*) fetal bovine serum and 300 μg/mL neomycin (G418). All cell lines were limited and cultured for no more than 30 passages, mycoplasma contamination was checked every other month, and all cell lines were verified to be negative for mycoplasma species using the MycoAlert Mycoplasma Detection Kit (Lonza, Houston, TX, USA). Unique signature identification of all osteosarcoma cells was confirmed by short tandem repeat (STR) DNA microsatellite fingerprinting analysis carried out at the CCSG CCLC core facility in The University of Texas MD Anderson Cancer Center (Houston, TX, USA).

Pramlintide was purchased from AstraZeneca Pharmaceuticals (Gaithersburg, MD, USA) as an injectable solution, SymlinPen 120 (1 mg/mL Pramlintide acetate), and 0.2 M sodium acetate buffer (pH 4.0) was used as the placebo control. For in vitro cell culture treatment, Pramlintide and the placebo were diluted with PBS before being added to the culture medium. For in vivo treatment, 30 µL (30 µg) of Pramlintide was directly injected intratumorally to the treatment mice; 30 µL of placebo control solution was injected in the same way to the control mice.

### 2.2. Western Blotting

Cell lysates were prepared with radioimmunoprecipitation assay (RIPA) buffer in the presence of proteinase inhibitor cocktail (Roche, Basel, Switzerland) on ice. Equal amounts of protein lysates (40 µg each) were separated by SDS-PAGE and then electrotransferred to nitrocellulose membranes (Invitrogen, Waltham, MA, USA). The membranes were blocked with TBS-T (tris-buffered saline and 0.5% Tween 20) solution containing 5% non-fat dry milk for 1 h and probed with anti-TAp63 (1:1000; BioLegend, San Diego, CA, USA), anti-TAp73 (1:500; IMG-246, Imgenex), anti-p73 (1:1000; human EP436Y; Abcam, Cambridge, UK), anti-p53 (WT, Vector Labs, Newark, CA, USA), anti-IAPP (1:1000, ab103580; Abcam), anti-calcitonin receptor (Abcam), anti-RAMP3 (Santa Cruz), anti–cleaved caspase-3, anti–cleaved PARP, anti-p21, and anti-p27 (Cell Signaling, Danvers, MA, USA) or, as a control, β-actin (Sigma-Aldrich, St. Louis, MO, USA) antibodies at 4 °C overnight. The membranes were washed 3 times with TBS-T and then incubated with the appropriate horseradish peroxidase-conjugated anti-mouse or anti-rabbit secondary antibody (Santa Cruz Biotechnology, Dallas, TX, USA) for 1 h and washed 3 times with TBS-T again. The specific protein was detected by enhanced chemiluminescence using the ECL Plus Western Blotting Detection Kit (GE Healthcare, Chicago, IL, USA). Densitometry analysis was performed, and the values were normalized with β-actin loading control.

### 2.3. Proliferation and Viability Analysis

Osteosarcoma cells were seeded in 24-well plates triplicated with full culture medium and treated with control PBS or Pramlintide diluted to 20 µg/mL with PBS. At 3, 5, and 7 days after treatment, culture mediums were collected well by well individually, and cells were then trypsinized and collected together with each matched wells’ culture medium. All collected cells, including their medium, were mixed with trypan blue staining solution (1:1) and loaded on a Vi-cell analyzer (Beckman Coulter, Brea, CA, USA); live cells were quantified for proliferation, and all cells that were stained with trypan blue were counted as dead cells for viability analysis.

### 2.4. Glycolytic Capacity Determination

Extracellular acidification rate (ECAR) was measured using the extracellular flux analyzer (SeaHorse Bioscience XF96, Agilent Technologies, Lexington, MA, USA) following the manufacturer’s instructions. Briefly, osteosarcoma cells were plated at a density of 1 × 10^4^ cells per well in the XF 96-well cell culture plates. Twenty-four hours after seeding, the culture medium was replaced with 180 μL of running medium and incubated for 1 h at 37 °C in a non-CO_2_ incubator. Before calibration, 20 μL of 50 mM glucose, 11 μM oligomycin, and 650 mM 2-DG were aliquoted into each port in the sensor cartridge. ECAR was measured after the addition of glucose and oligomycin but before the addition of 2-DG. Extracellular acidification rate was normalized to mpH/min.

### 2.5. Flow Cytometry Analysis of Apoptosis

Cultured cells were washed with 1× PBS once, trypsinized, and resuspended in 1× PBS containing 0.5% fetal bovine serum. Cells were stained with PE Annexin V Apoptosis Detection Kit (BD Pharmingen, Franklin Lakes, NJ, USA) for 30 min at 4 °C in the dark and examined with a fluorescence-activated cell-sorting flow cytometer (BD Biosciences, San Diego, CA, USA); PE staining positive cells were gated as apoptotic cells.

### 2.6. In Vivo Animal Experiments

Four- to five-week-old Nude (Nu/Nu) mice were purchased from the National Cancer Institute. The mice were maintained in a specific pathogen-free animal facility approved by the American Association for Accreditation of Laboratory Animal Care. The animal experiment protocol was approved by the Institutional Animal Care and Use Committee of MD Anderson Cancer Center (00000896-RN02, Gene- and Chemotherapy of Sarcoma Pulmonary Metastases) for Dr. Kleinerman. CCH-OS-D-luc cells in the mid-log-growth phase were harvested by trypsinization and resuspended in PBS and stored together with Matrigel (HC, Phenol-Red Free, LDEV-free, Corning Life Sciences, Corning, NY, USA) on ice before injection. A total of 0.5 million CCH-OS-D-luc cells mixed with Matrigel were subcutaneously injected on both sides of the abdomen area of the Nude mice; 5 days later, an approximately 3 to 5 mm diameter tumor at each injection site was formed. The bioluminescence imaging of the tumor was taken before starting the treatment. Tumors on both sides were treated by Intratumoral injection of 30 µg of Pramlintide twice a week, and imaging was taken every week. Bioluminescence imaging was taken by IVIS Spectrum System using Live Image software (4.5.5).

### 2.7. TUNEL Staining

TUNEL staining was performed using the DeadEnd Fluorometric TUNEL System (G3250, Promega, Madison, WI, USA). Briefly, fresh frozen sections (5 µm) were air dried at room temperature for 40 min, fixed in cool acetone for 10 min, and permeabilized by proteinase K solution (20 µg/mL) for 10 min. After repeat fixing for 5 min and being equilibrated with equilibration buffer for 10 min, the sections were labeled with TdT reaction mix at 37 °C in a humidified chamber for 60 min, avoiding light exposure. The reaction was stopped for 15 min by 2× SSC buffer, then DAPI nuclear stain in mounting medium was added. Two to four tumor areas (hot spots) from each section were imaged at 200× by Leica DMi8 microscopy. Quantification of TUNEL was performed by Image J software to count the number of TUNEL cells and total cells in the same field. This measurement was expressed as the percentage of TUNEL cells in the field.

### 2.8. Ki67, CD31, and HIF1-α Immunofluorescence Staining

Based on the corresponding hematoxylin and eosin staining, 10 fresh frozen tumor sections (5 µm) were selected from control and pramlintide treatment groups to do Ki67, CD31, and HIF1-α immunofluorescence staining. The sections were air dried at room temperature for 40 min, fixed in cool acetone for 10 min, and permeabilized by 0.25% TritonX-100 for 5 min. Endogenous peroxidase activity was blocked by incubating the sections with 3% hydrogen peroxide for 10 min. After three PBS washes, the sections were blocked by 10% normal goat serum for 1 h at room temperature in a humid chamber box, then incubated with Ki67 (1:200), CD31 (1:100), HIF1-α (1:100), CD86 (1:200), and CD163 (1:200) primary antibodies (Ki67, Invitrogen, MA5-14520; CD31, Abcam, ab28364; HIF1-α, Invitrogen, PA1-16601, CD86, NeoBiotechnologies, 942-RBM4-P0; CD163, Abcam, ab182422). Samples were stored at 4 °C overnight with a parafilm coverslip in a humid chamber box to prevent evaporation. Then, the sections were incubated with a secondary antibody (1:500, Alexa Fluor 488 goat anti-rabbit, Invitrogen, A11034) at room temperature for 1 h and counterstained with DAPI in mounting medium. Two to four tumor areas (hot spots) from each section were imaged at 200× by Leica DMi8 microscopy. Quantification of Ki-67 was performed by Image J software to count the number of Ki67-positive cells and total cells same field. This measurement was expressed as the percentage of Ki67-positive cells. Quantification of CD31 and HIF1-α were performed by Image J software to measure the fluorescence intensity of CD31 and HIF1-α staining and expressed as relative fluorescence units (RFUs).

### 2.9. Statistical Analysis

All values were reported as means ± SEMs. A two-tailed Student *t*-test was used to statistically evaluate the experimental results. One-way ANOVA was used for statistical analysis of animal experiment results. *p* values less than 0.05 were considered statistically significant.

## 3. Results

### 3.1. Osteosarcoma Cells with p53 Alterations Express High Levels of the Dominant-Negative (DN/ΔN) Isoforms of p63 and p73 and a Low Level of the Transactivation Domain (TA)p63 and Tap73 Isoforms

Pramlintide has been shown to induce tumor regression in p53-deficient thymic lymphomas [22,23], indicating that this agent may have activity in other tumors with p53-alterations such as osteosarcoma. To confirm the alteration of p53 in several different osteosarcoma cell lines, we performed a Western blot analysis of p53 using total cell lysates (Figure 1A). CCH-OS-D, MG63.2, LM7, and U2OS cells showed low or no p53 expression (similar to the human non-small cell lung carcinoma H1299 cells) compared to normal p53 expression in hFOB cells. CCH-OS-O cells had mutated p53 as previously described [28]. SJSA-1 cells overexpressed murine double minute 2 (MDM2), which inactivates p53 [29] and therefore leads to resistance to p53-dependent apoptosis [30].

The p53 family members p63 and p73 are expressed as two N-terminal splice variants. These full-length acidic transactivation domain (TA) isoforms are structurally and functionally similar to wild-type p53. The ΔNp63 and ΔNp73 isoforms lack the acidic transactivation domain, are often overexpressed in cancer cells, and act primarily in a dominant-negative fashion against all p53 family members [31]. These ΔN isoforms inhibit the tumor suppressive functions of p53, TAp63, and TAp73. The deletion of either ΔNp63 or ΔNp73 in p53-deficient mouse lymphomas resulted in tumor regression mediated by metabolic programming [22]. Western blot analysis of osteosarcoma cells exhibited expression of one or both of the ΔNp63 or ΔNp73 isoforms comparable to the expression pattern seen in the positive control H1299 cells [22] (Figure 1B,C). Normal osteoblastic cells (e.g., hFOB cells) express p53 predominantly, and therefore, as a negative feedback signal, express low levels of the TA isoforms of p63 and/or p73. Although ΔNp63 was seen in hFOB cells, these normal cells express the ΔN forms of p63 and/or p73 as positive feedback for the overall regulation balance of p53 family proteins (Appendix A).

Pramlintide-mediated tumor regression was shown to be mediated by metabolic reprogramming through the upregulation of IAPP [22]. The IAPP receptors consist of the calcitonin receptor (CalcR) and the co-receptor, receptor activity modifying protein 3 (RAMP3), which controls IAPP’s specificity to CalcR [32,33]. IAPP functions through both CalcR and RAMP3 to inhibit glycolysis and induce apoptosis. Therefore, expression of these receptors in osteosarcoma cells is essential for the therapeutic activity of Pramlintide. Western blot analysis demonstrated that both receptors were integrated and expressed in all osteosarcoma cells as well as in normal hFOB cells (Figure 1D,E).

Elevated levels of ΔNp63 and ΔNp73 in p53-deficient thymic lymphoma cells repressed IAPP expression. Silencing of either ΔNp63 or ΔNp73 restored the IAPP levels, inhibiting glycolysis and inducing tumor cell apoptosis [22]. Similar to what was seen in H1299 cells, Western blot analysis demonstrated that IAPP expression in osteosarcoma cells was decreased compared to the hFOB positive control cells (Figure 1F). These data are in line with the investigation by Venkatanarayan et al., showing that ΔNp63/ΔNp73 blocks the ability of TAp53, p63, and p73 to bind to the IAPP promoter and thereafter resulting in decreased expression of endogenous IAPP. These data support that elevated expression of ΔNp63 and ΔNp73 inhibits IAPP expression in osteosarcoma cells and that restoring IAPP using Pramlintide may lead to metabolic reprogramming and induction of apoptosis.

### 3.2. Pramlintide Inhibits Osteosarcoma Cell Glycolysis and Proliferation, Inducing Cell Cycle Arrest and Apoptosis

Tumor cell response to Pramlintide treatment has been shown to be dependent on glycolysis and the cell’s glycolytic capacity (GC) [22,34,35]. We therefore evaluated the metabolic profile of six osteosarcoma cell lines with ΔNp63 and ΔNp73 by assessing their extracellular acidification rate (ECAR) using a Seahorse XF96 analyzer (Figure 2A). We demonstrated different GCs in the osteosarcoma cell lines. CCH-OS-D, CCH-OS-O, and U2OS showed high GC, MG63.2 cells showed intermediate GC, and LM7 and SJSA-1 cells showed low GC. We selected osteosarcoma cells with high GC (CCH-OS-D) and middle range GC (MG63.2) to treat with two different doses (15 µg/mL and 20 µg/mL) of Pramlintide to evaluate the drug’s effect on glycolysis. Pramlintide reduced the GC of both the CCH-OS-D (Figure 2B) and MG63.2 (Figure 2C) cells in a dose-dependent manner. The reduction of GC was more pronounced in the CCH-OS-D cells than in the MG63.2 cells. In contrast, there was no significant reduction of GC in the cells that had low GC (LM7 and SJSA-1; data not shown). To determine whether the inhibition of glycolysis affected cell proliferation, CCH-OS-D (high GC), MG63.2 (mid GC), and LM7 (low GC) cells were treated with either PBS or Pramlintide at 20 µg/mL. Cell proliferation was then quantified after 3, 4, or 5 days. Pramlintide treatment significantly decreased tumor cell proliferation in all three cell lines (Figure 2D–F).

Blocking cell glycolysis also induced apoptosis in CCH-OS-D and MG63.2 cells as quantified by Annexin staining (Figure 2G,H). As anticipated, since LM7 cells have a low GC, Pramlintide inhibited LM7 cell proliferation but did not induce apoptosis (Figure 2I). Table 1 shows the correlation of the basal level of glycolysis with Pramlintide-induced growth inhibition and apoptosis.

### 3.3. Effect of Pramlintide on Tumor Growth In Vivo

Having demonstrated maximum Pramlintide activity in vitro with CCH-OS-D cells, we selected these cells to determine whether Pramlintide is effective in vivo. A subcutaneous tumor mouse model using luciferase-labeled CCH-OS-D cells (CCH-OS-D-luc) was used to evaluate Pramlintide efficacy in vivo. CCH-OS-D-luc cells mixed with Matrigel were injected subcutaneously on each side of the abdomen. When the tumor diameter reached 5 mm (~5 days), bioluminescence imaging of the tumor was done before starting Pramlintide treatment. Tumors on both sides were treated by an Intratumoral injection of 30 µg of Pramlintide or PBS twice a week for 3 weeks. Tumor response on both sides was determined by imaging (Figure 3A). Prior to therapy, there was no difference in tumor size. Tumors in the control group increased in size after 26 days, while those in the Pramlintide-treated group showed a significant decrease in size after 26 days (Figure 3B,C and Appendix A). Pramlintide-treated tumors on day 26 showed a 66% reduction in size compared to pretreatment values and a 75% reduction compared with the control tumors.

### 3.4. Pramlintide Treatment Induces Tumor Cell Apoptosis and Reduces Tumor Proliferation and Hypoxia In Vivo

Pramlintide treatment of CCH-OS-D cells in vitro induced apoptosis and cell cycle arrest as confirmed by the induction of 2 apoptosis markers (cleaved caspase-3 and PARP; Figure 4A,B) and the expression of the cell cycle arrest markers p21 and p27 (Figure 4C,D), with a decrease in cyclin D3, the downstream target (Figure 4E). We therefore evaluated the treated and untreated tumors for apoptosis (TUNEL), proliferation (Ki-67), and hypoxia (HIF-1α) at the end of therapy. There was a significant increase in tumor cell apoptosis and a decrease in proliferation and hypoxia in the Pramlintide-treated tumors compared to the controls (Figure 5A–F). In contrast, we found no difference in tumor vessels (CD31 staining) (Figure 5G,H) or micro vessel density (Figure 5I) between the control and Pramlintide-treated tumors. We also found no difference in tumor macrophage M1 (CD86 staining) and M2 (CD163 staining) populations between the control and Pramlintide-treated tumors (Appendix A), suggesting that Pramlintide-induced antitumor activity in this model was via a direct effect on the tumor cells and not mediated by immune cells or an effect on the tumor vascularity.

Human CCH-OS-D cells require the use of immune-deficient mice that lack T-cells and NK cells (with normal macrophage function). Therefore, as expected, no T-cells or NK cells were detected in either the control or Pramlintide-treated tumors. Taken together, these data indicate that Pramlintide increased tumor cell apoptosis and decreased cell proliferation and hypoxia by inhibiting the osteosarcoma cell metabolic process, resulting in tumor regression.

## 4. Discussion

The full-length homolog isoforms of p53 (TAp63 and TAp73) behave similarly to p53 and can be substituted for the missing or mutated p53. These homologs can bind to and activate the transcription of p53-responsive promoters, inducing cell cycle arrest and apoptosis in response to cellular stress. However, alternative splicing variants produce truncated ΔN isoforms that are transcriptionally inactive. Consequently, ΔNp63 and ΔNp73 act as inhibitors of p53 and other active family members. Both ΔN forms also inhibit the expression of IAPP, thereby promoting glucose uptake and cell glycolysis. Restoring IAPP in p53-deficient cells that express ΔNp63 or ΔNp73 blocked cell glycolysis, subsequently inducing cell cycle arrest and apoptosis [22,23].

The data presented here show for the first time that, in addition to abnormalities in p53, osteosarcoma cells express ΔNp63 and ΔNp73. We also demonstrated that IAPP is downregulated in osteosarcoma cells and that treatment of these cells with Pramlintide, an FDA-approved analogue of IAPP used in the treatment of type 2 diabetes, altered the metabolic profiles of osteosarcoma cells with high or mid-range GC but had no effect on cells with low GC. Decreasing the GC of high and mid-range cells resulted in decreased proliferation and apoptosis as defined by annexin V staining, caspase-3, and PARP cleavage, upregulation of p21 and p27, and decreased cyclin D3. In contrast, Pramlintide failed to induce apoptosis in the cells with low GC. Thus, Pramlintide efficacy correlated with the GC of the cells. High cancer cell glucose consumption is often seen in osteosarcoma patients using PET-CT scans with 18FDG glucose in both osteosarcoma primary tumors and metastatic deposits.

We went on to investigate whether Pramlintide had anti-tumor activity in vivo. Intratumoral injection of Pramlintide inhibited tumor growth after six treatments. Treated tumors showed increased apoptosis and decreased cellular proliferation as well as decreased hypoxia. Taken together, these data indicate that Pramlintide treatment altered the tumor microenvironment (TME).

Cancer cells, including osteosarcoma cells, have increased rates of glycolysis and glucose consumption to compensate for the increased ATP needed for tumor growth [36]. Decreasing available glucose through the use of Pramlintide can therefore decrease glucose availability, starve the cells, and decrease proliferation, which in turn will inhibit the growth rate of the tumor. While we did not investigate whether Pramlintide affected metastatic spread to the lungs or the growth of lung metastases, this is a beginning step and substantiates the principle that targeting the metabolic network using this agent is feasible and has therapeutic potential.

Pramlintide is a synthetic peptide with a short half-life (<1 h) when given subcutaneously. Type 2 diabetes patients are treated with Pramlintide 60–120 micrograms s.c. 3×/day before meals. Our results were achieved using a 2×/week intratumoral schedule. Schedule, route, and best dose response remain to be determined. Furthermore, direct comparison with other agents known to decrease glucose (e.g., metformin) could help to identify the optimal approach against osteosarcoma.

Our data also suggest that combining Pramlintide with other forms of therapy may be beneficial. For example, the lactate converted during glycolysis is transported out of the cancer cells. The increased rates of glycolysis and glucose consumption create an acidic TME with increased lactic acid that impairs anti-tumor immunity by inhibiting the activity of activated T-cells, including CAR T-cells, and promoting T-reg cells [37,38,39]. This metabolically immunosuppressive TME induces T-cell hyperresponsiveness. T-cell function, proliferation, and differentiation are also blunted in vivo in response to high levels of lactic acid [40,41].

In addition, hypoxia in the TME has been associated with evasion of immune detection, immune escape, and immune cell apoptosis [42]. HIF-1α specifically has been shown to inhibit immune cell function within the TME [43]. This disruption allows tumors to escape immune-mediated killing, thus decreasing the efficacy of numerous immunotherapeutic approaches, including anti–PDL-1 and anti–CTLA4 [42]. Conversely, inhibition of HIF-1α has been shown to enhance the anti-tumor activity of T-cells. Inhibiting glycolysis and decreasing HIF-1α in the TME was also shown to increase the generation of memory T-cells and improve anti-tumor functionality [44]. We demonstrated that Pramlintide therapy resulted in decreased HIF-1α in the TME, again suggesting that Pramlintide can help alter the TME from one that is immunosuppressive to one that supports immune cell mediated anti-tumor activity.

We were not able to assess changes in T-cell content or sub-type of T-cells within the tumor, as our in vivo studies were performed in immunodeficient mice that lacked T- and NK cells. Ascertaining whether Pramlintide enhances T-cell-mediated immune therapy will therefore require the use of an immunocompetent mouse osteosarcoma model. We were unable to detect a change in macrophage numbers or macrophage phenotype in the treated tumors.

In summary, the data presented suggest that altering the metabolic immunosuppressive TME by inhibiting glycolysis using Pramlintide, an FDA-approved drug currently used to treat type 2 diabetes, represents a novel strategy for treating osteosarcoma, particularly in conjunction with various immunotherapies that rely on T-cell activation and anti-tumor function. Immunotherapy has been largely ineffective in patients with osteosarcoma metastases. This has been linked to poor penetration of T-cells into the tumor and T-cell exhaustion. Since the efficacy of immunotherapy that relies on T-cells to mediate the anti-tumor response is dampened by both an acidic and hypoxic TME, our data offer a way to improve tumor response. The fact that Pramlintide is already FDA-approved makes it particularly attractive for potential translation into clinical trials for patients with relapsed osteosarcoma with few therapeutic alternatives.

## 5. Conclusions

We demonstrated for the first time that osteosarcoma cells express ΔNp63 and ΔNp73 in addition to abnormalities of p53. Both ΔN forms inhibit the expression of IAPP, thereby promoting glucose uptake and cell glycolysis. Restoring IAPP using Pramlintide (an analogue of amylin) has been shown to block cell glycolysis, resulting in cell cycle arrest and apoptosis. Our data show that Pramlintide treatment in vitro inhibited the glycolytic metabolism of osteosarcoma cells with high or mid-range GC, leading to a decrease in cell proliferation and apoptosis. Similarly, Intratumoral Pramlintide therapy induced tumor regression with an increase in several apoptosis markers, confirming tumor cell death. Tumor hypoxia also decreased. This is the first report showing the potential efficacy of Pramlintide in osteosarcoma tumor regression. These results indicate that Pramlintide, with its ability to modulate osteosarcoma cell metabolic programming, may be a novel therapeutic approach for patients with relapsed osteosarcoma, particularly in combination with immunotherapies involving T-cell-mediated cytotoxicity since both hypoxia and a glucose-depleted TME compromise T-cell function and proliferation.

## Figures and Tables

**Figure 1 cancers-14-04310-f001:**
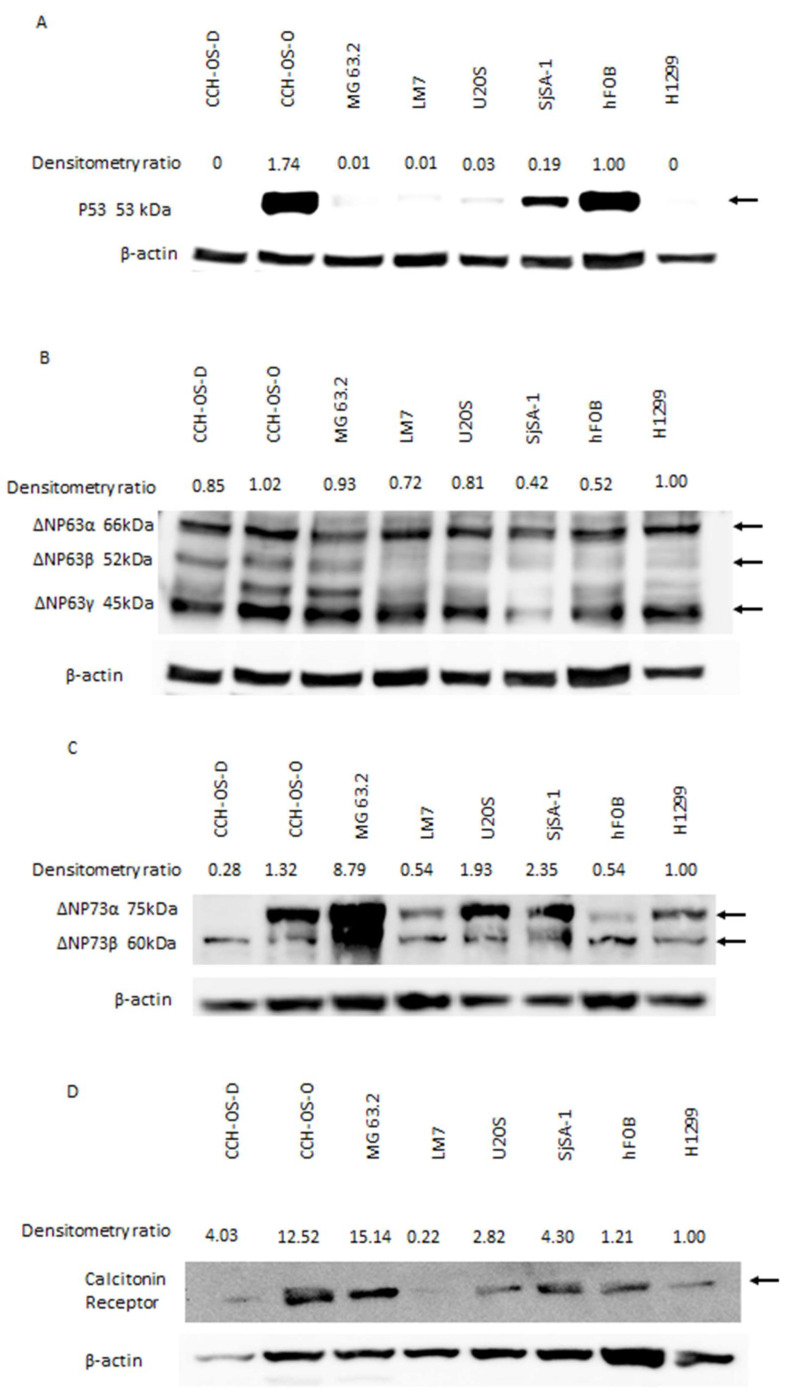
Osteosarcoma cells express high levels of ΔN isoforms of p63/73 and low levels of TA isoforms. Western blot analysis. (**A**) p53 expression. Positive control was human fetal osteoblastic cells (hFOB); negative control was H1299 cells. (**B**) ΔNp63 and (**C**) ΔNp73 expression in osteosarcoma cells and H1299 cells (positive control). (**D**) Amylin receptor CalcR and (**E**) RAMP3 expression. (**F**) Endogenous IAPP (amylin) expression. Lysates from hFOB and H1299 cells were used as the positive and negative controls, respectively. Full Western blot images can be found at File S1.

**Figure 2 cancers-14-04310-f002:**
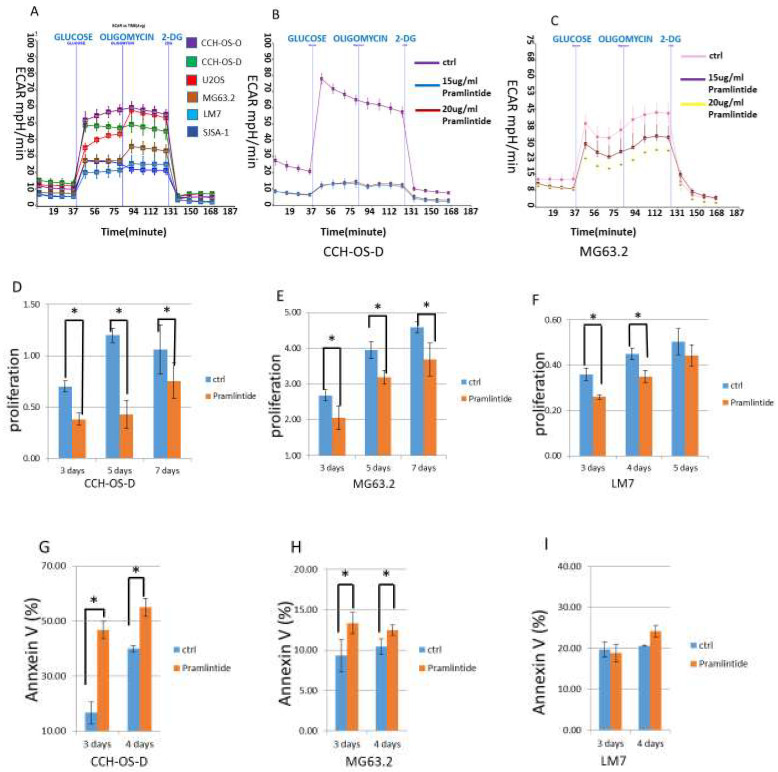
Effect of Pramlintide on cell glycolysis, proliferation, and apoptosis. (**A**) The metabolic profiles of six osteosarcoma cells lines were evaluatd by assessing the extracellular acidification rate (ECAR) using Seahorse Bioanalyzer XF96 analysis. CCH-OS-D (**B**) and MG63.2 (**C**) cells showed different glycolytic responses to Pramlintide treatment in a dose-dependent manner. (**D**–**F**) Effect of Pramlintide (20 µg/mL) over time on cell proliferation as quantified using trypan blue staining and the Vi-cell analyzer. (**G**–**I**) CCH-OS-D, MG63.2, and LM7 cells were treated with PBS or Pramlintide (20 µg/mL) for 3 or 4 days. Apoptosis was then quantified using Annexin V-PE staining and flow cytometry. Bars represent standard deviation. * *p* < 0.05.

**Figure 3 cancers-14-04310-f003:**
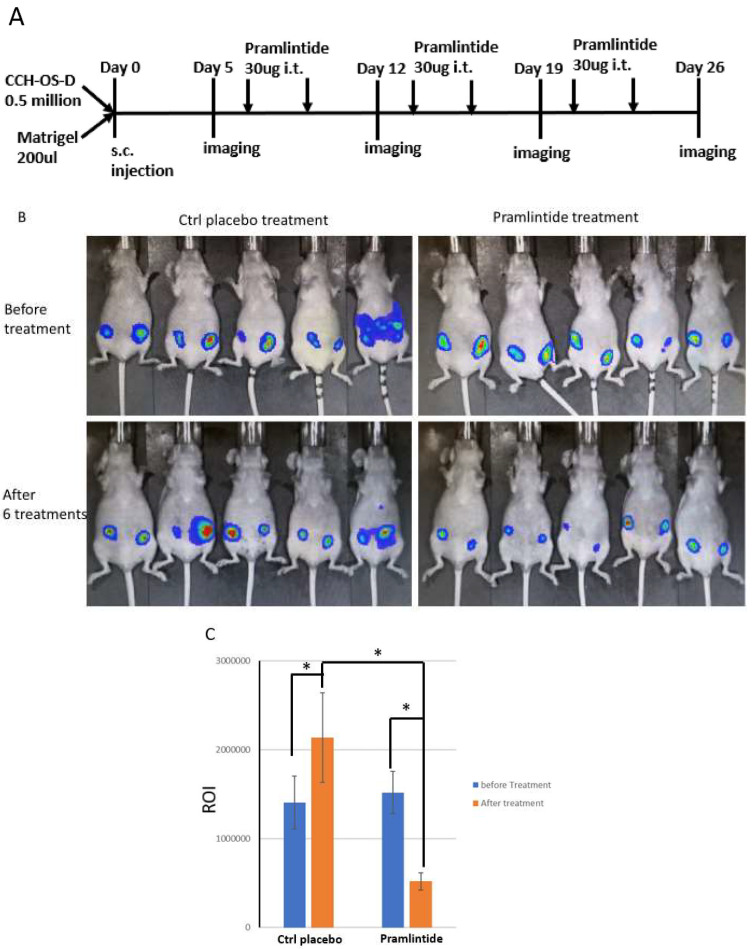
Pramlintide treatment inhibited osteosarcoma tumor growth in vivo. (**A**) Schematic diagram of Pramlintide treatment. (**B**) Bioluminescence imaging of the tumors before treatment and 3 days after the final, sixth treatment. (**C**) Quantification of bioluminescence imaging before and after control placebo or Pramlintide therapy. * *p* < 0.05.

**Figure 4 cancers-14-04310-f004:**
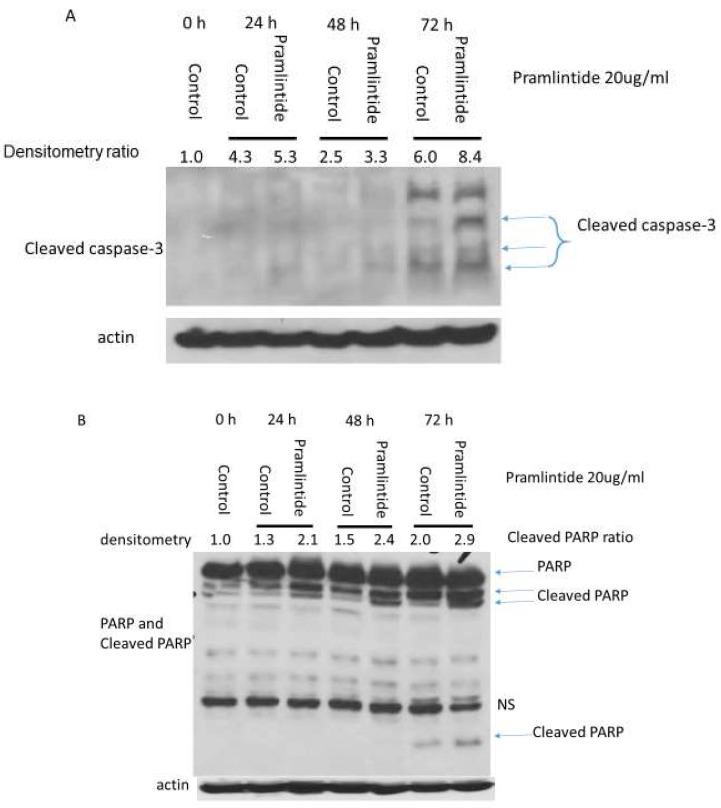
Pramlintide treatment in vitro induced cell cycle arrest and apoptosis. Western blot analysis for cleaved caspase-3 (**A**), cleaved PARP (**B**), p21 (**C**), p27 (**D**), and cyclin D3 (**E**) following Pramlintide treatment. Full Western blot images can be found at File S1.

**Figure 5 cancers-14-04310-f005:**
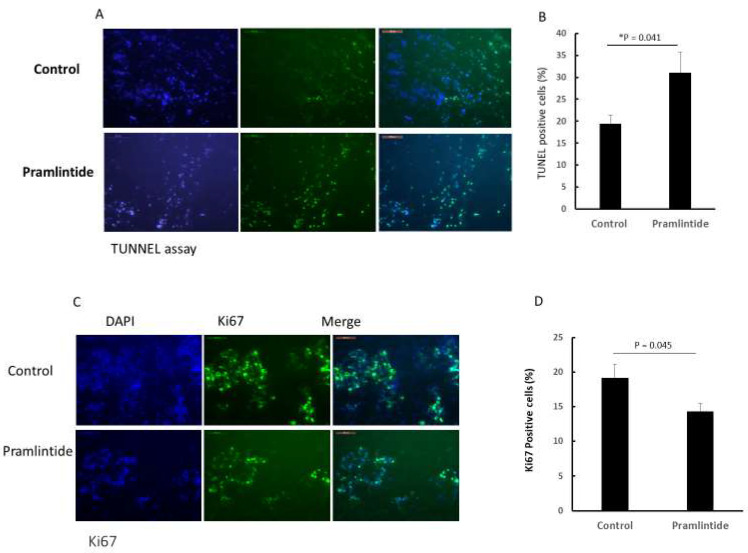
Pramlintide treatment in vivo induces tumor apoptosis and decreases tumor proliferation and hypoxia. Tumors were harvested following therapy and analyzed for apoptosis, proliferation, and hypoxia using TUNEL (**A**,**B**), KI-67 (**C**,**D**), and HIF-1α (**E**,**F**), respectively. Tumor vessels were assessed using CD31 (**G**–**I**). Scale bar: (**A**) = 50 µm, (**C**,**E**,**G**) = 100 µm. Quantification of fluorescent staining (**A**,**C**,**E**,**G**) was performed by analyzing four random tumor areas using ImageJ software to count the number of positive cells (**B**,**D**) and positive fluorescence intensity in each field (**F**,**H**). Bars represent standard deviation. * *p* < 0.05. (**I**) Micro vessel density was analyzed by quantifying the number of CD31+ cells per field.

**Table 1 cancers-14-04310-t001:** Correlation of basal glycolysis with Pramlintide efficacy.

Cell Line	Basal Levels of Glycolysis	Pramlintide Treatment		
		Glycolysis Suppression	Cell Growth Inhibition	Apoptosis
CCH-OS-D	+++	++++	+++	+++
MG63.2	++	++	++	++
LM7	+	+/−	+/−	−

++++ highest response, +++ high response, ++ medium response, + low response, − no reponse.

## Data Availability

The data presented in this study are available on request from the corresponding author.

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
