# Peer review of "Pramlintide: A Novel Therapeutic Approach for Osteosarcoma through Metabolic Reprogramming"

_cancers, 2022, doi:10.3390/cancers14174310_

Round 1

Reviewer 1 Report

The authors have reported here that pramlintide, an FDA approved drug for type 2 diabetes, could be repurposed to be an anti-tumor molecule. Mechanistically, pramlintide interferes with glycolysis pathway in osteosarcoma tumor cells, resulting in decreased cell growth and increased cell death. These bona fide anti-tumor effects were also observed in xenograft mouse model in which tumor bearing mice were intratumorally administrated with pramlintide. This data indicates a potential efficacy of the anti-diabetic pramlintide against osteosarcoma through targeting glycolysis in tumor. However, the systemic side effects of pramlintide on glucose homeostasis need to be determined to clarify the drug benefit is driven by cell autonomous mechanisms instead of by the systemic regulation.

Special concerns:

1. Structural reorganization of all figures will be great. The resolution also looks dim. The labels are hard to read (Fig.2), what were injected in each portal in the ECAR assays?

2.  Fig.1, The WB for the TA p63 and p73 isoforms is missing. Does high expression of ΔNp63 and ΔNp73 really correlate with expression of IAPP receptors amongst cell lines? If not, the statement “these data suggest that elevated expression of ΔNp63 and ΔNp73 inhibits IAPP expression in osteosarcoma cells” lacks convincing evidence.

3. Fig.2, LM7 cells have low glycolytic capacity and has no response to pramlintide in ECAR assay. Why is the cell proliferation of this line still affected by the pramlintide? Secondly, does the GC correlated to the expression of IAPP receptors amongst cell lines?

4. Fig.3, do you have non-tumor control mice? The pramlintide administration is supposed to reduce glucose levels per se. Do you measure glucose levels one day post the intratumoral injection? This is important to rule out the side-effects that the observed tumor regression may in fact come from the systemic glucose-lowering due to the drug itself.  

5. Fig.5, scale bar is missing. 

Reviewer 2 Report

Comments:

1.      What are the findings in the current report that provide new understanding of cancer biology, pathogenesis, diagnosis, or new treatment targets? Which of the new targets are important for clinical characterization and can these be analyzed using IHC? How does the current report advance our knowledge, and does it provide a “roadmap” for future systematic analyses and laboratory evaluation? If so, exactly what specifically is in this roadmap?

2.      Are there new therapeutic targets identified and are these same in other tumor lineages?

3.      Growing evidence suggests that cancer-initiating cells with metabolic plasticity can promote tumor progression and therapy resistance. What effect does this have on the current finding, which utilize pools of differentiated cancer cells?

4.      It will be intriguing to compare the bioenergetic feature of cancer-initiating cells and differentiated cells in current study to unveil novel treatment options.

5.      The author administers Pramlintide treatment via intratumoral injection. Why intratumoral as opposed to intravenous? Please justify rationally.

6.      Does the subcutaneous osteosarcoma model replicate osteosarcoma's clinically relevant metabolic microenvironment? Please justify rationally.

7.      Additionally, previous research has shown that different anatomic locations govern the different metabolic compositions of the same tumor cells. How does the author infer this based on their findings? Please justify rationally.

8.      In relation to the above comment, why didn’t the author utilize other anatomic locations of tumor injection such as intraosseous injection, which clinically mimic osteosarcoma? Please provide key data using intraosseous injection to correlate the findings.

9.      Is the metabolic composition of tumors influenced by tumor size, anatomical location, tumor tissue of origin, diet, and tumor genetics? Is so, what is the evidence?

10.  What is the effect of knocking down ΔNp63/73 on glycolysis, proliferation, and apoptosis with or without Pramlintide?

11.  Does CALCR and RAMP3 receptors knockdown attenuate Pramlintide effect on tumor growth? Please show the data.

12.  What effect does secretory IAPP have on cancer cell glycolysis and oxidative phosphorylation that may provide additional energy to sustain cell proliferation.

13.  How does the author respond to the previous study by Palorini et al. indicating that MG-63 relies on glutamine oxidation rather than glycolysis?

14.  What effect does Pramlintide have on cells cultured in glucose/glutamine-limited conditions? Show relevant data.

15.  What effect does lowering glucose/glutamine concentration have on the response to known chemotherapy drugs in combination with Pramlintide?

16.  In Figure 2A-C, author showed only glycolysis (ECAR) data. What about the respiration (OCR) data of the indicated cell line treated with or without Pramlintide that may provide additional energy to sustain cell proliferation?

17.  The author only demonstrated the mode of Pramlintide treatment on tumor cells but provides no evidence of the underlying mechanism by which Pramlintide inhibits glycolysis. Please provide experimental data to correlate the findings.

18.  Given the context of the discussion about tumors producing high levels of glucose, it is essential that the authors show the lactate data. This finding, I believe, will be of interest to readers because it affects how we think about glucose handling in these tumors.

19.  The author states in result section 3.2 that LM7 cells have a low GC, so Pramlintide inhibited LM7 cell proliferation but did not induce apoptosis. What conclusions does the author draw from the observation that hypoxic tumors have altered nutrient levels, including glucose? Please provide the tumor metabolomics data to rule out this discrepancy.

20.  What effect does Pramlintide treatment have on mouse body weight? Please provide data, as previous reports have revealed suppressed food intake and body weight. Also, how does this affect the current findings' conclusion?

21.  How is Pramlintide in vitro and in vivo dose rationalized?

22.  No information is provided on how sample size calculation is performed making it difficult to assess statistical validity.

23.  Furthermore, how can the author claim that Pramlintide treatment altered the tumor microenvironment by using an immune-compromised nude mouse model? (TME). The author should be cautious regarding the experiment interpretation and is advised not to overstate/extrapolate the findings without meticulously executing the experiments.

24.  The resolution of the IF image is diffuse and not clear enough to interpret the findings. Please provide better-resolution images with quantification.

25.  Furthermore, TUNNEL, Ki67, and HIF-1 staining appear to show no difference between control and Pramlintide treatment. Please rule out this discrepancy.

26.  During the discussion, similar claims were made without conducting the experiment. Please avoid overstatement and proceed with caution.

27.  Please change the word microphage to macrophage.

28.  Please provide the mice survival data to correlate the findings clinically.

29.  The supporting data is also not available to review, making it impossible to identify how robust it is.

30.  In all seahorse data, please highlight the axis. It is difficult to keep track of the time and the ECAR value.

Reviewer 3 Report

List of authors- Research in superscript of fist author affiliation probably a typo

Simple summary  -recommend adding   "often"  so Osteosarcoma cells often express  (doubt they always express abnormal p53, p63, p73).

Introduction: suggest adding a simple figure to explain your approach figure to explain

Results are convincing for  high glucose metabolizing tumors. An Oncologist would think of PET-CT as the clinical correlate so suggest adding to paragraph of discussion after  Thus, pramlintide efficacy correlated with the GC of the cells. " High cancer cell glucose consumption is often seen in osteosarcoma patients using PET-CT scans with 18FDG glucose in both osteosarcoma primary tumors and metastatic deposits. "

in Discussion page 15 second paragraph need to put your findings into context.

"Pramlintide is a synthetic peptide with short half-life (<1 hour) when given sc. Type 2 diabetes patients get Pramlintide 60-120 micrograms sc 3x/day before meals. Our results were achieved using a 2x/week intratumoral schedule. Schedule, route and best dose response remain to be determined. Furthermore direct comparison with other agents known to decrease glucose (e.g. metformin) could help to know best approach against osteosarcoma, too.

Round 2

Reviewer 2 Report

Overall, the authors should be commended for their efforts in responding to the comments, but I'm concerned about the low novelty that has been lost because of prior significant publications by (Venkatanarayan et al.), as the author pointed out, which amply demonstrated mechanistically Pramlintide mode of action in the p53-deficient model. I believe the key message is still difficult to pull out from the manuscript and suggest a brief statement that more clearly demonstrates its novelty rather than saying… This is the first report showing the potential efficacy of Pramlintide against osteosarcoma.   

Author Response

Response to Reviewer #2 round 2 Comments:

We thank the reviewer for acknowledging our concerted efforts to respond to the comments. With regard to the comment on novelty and low significance, we respectfully submit that we have addressed this in the introduction and the discussion of the manuscript.  However, here is a more extensive explanation.

The survival rate for patients with osteosarcoma has not improved in >25 years despite the use of multiple different chemotherapy combinations and still hovers around 60-65% (1-3). Immunotherapy as well has not been shown to improve patient outcome.  A new approach is clearly needed if we are break through this plateau.  While our data are in keeping with the findings of Venkatanarayan et al., those studies were done in thymoma, an extremely rare tumor in children and adolescents.  One can not assume that what works in thymoma is applicable in osteosarcoma.  Indeed, the significant activity of check-point inhibitors in lung cancer and other adult cancers has not proven effective in patients with sarcoma.  Chemotherapy regimens themselves are not the same in patients with different types of solid tumors.  What is effective in children with lymphoma and brain tumors, and the treatment regimens used, are totally different than the standard of care for OS.  Therefore, prior to translating the use of Pramlintide into a clinical trial for children and adolescents with OS, one must demonstrate that it has the potential for activity against OS.  Otherwise there will be no justification for giving a drug to these patients that has not shown preclinical activity in the relevant tumor type and there will never be a justification for moving forward. Our data provide this needed preclinical data which is intended to stimulate interest in its use as we strive to find a place for immunotherapy which so far has shown little effectiveness.

We respectfully think that we have made this clear.  We do not think it necessary to lengthen the discussion and restate what we outlined in the introduction and the discussion. We hope that the above has provided the needed explanation to this reviewer that our studies are novel and significant as they pertain to the treatment of osteosarcoma.

  1. Lancet Oncology 17:1396-1408, 2016)
  2. Journal of Clinical Onc (26:633-638)
  3. J. Clinical Onc 20:776-790).

Reviewer 3 Report

For clinicians and patients with osteosarcoma, this provides scientific evidence to support metabolic treatment of some osteosarcomas in which glycolysis is overly active. The in vitro and in vivo results and conclusions are clearly presented.

Author Response

Response to Reviewer #3 round 2 Comments:

We total agree reviewer's comments, "For clinicians and patients with osteosarcoma, this provides scientific evidence to support metabolic treatment of some osteosarcomas in which glycolysis is overly active. "